# Evaluation of Two Broadly Used Commercial Methods for Detection of Respiratory Viruses with a Recently Added New Target for Detection of SARS-CoV-2

**DOI:** 10.3390/v14071530

**Published:** 2022-07-13

**Authors:** Monika Jevšnik Virant, Tina Uršič, Rok Kogoj, Miša Korva, Miroslav Petrovec, Tatjana Avšič-Županc

**Affiliations:** Institute of Microbiology and Immunology, Faculty of Medicine, University of Ljubljana, Zaloška 4, 1000 Ljubljana, Slovenia; monika.jevsnik@mf.uni-lj.si (M.J.V.); tina.ursic@mf.uni-lj.si (T.U.); rok.kogoj@mf.uni-lj.si (R.K.); misa.korva@mf.uni-lj.si (M.K.); mirc.petrovec@mf.uni-lj.si (M.P.)

**Keywords:** SARS-CoV-2, COVID-19, respiratory viruses, validation, molecular assay

## Abstract

The clinical symptoms caused by the severe acute respiratory syndrome coronavirus 2 (SARS-CoV-2) are nonspecific and can be associated with most other respiratory viruses that cause acute respiratory tract infections (ARI). Because the clinical differentiation of COVID-19 patients from those with other respiratory viruses is difficult, the evaluation of automated methods to detect important respiratory viruses together with SARS-CoV-2 seems necessary. Therefore, this study compares two molecular assays for the detection of respiratory viruses, including SARS-CoV-2: the Respiratory Viruses 16-Well Assay (AusDiagnostics, Pty Ltd., Mascot, Australia) and the Allplex™ RV Essential Assay coupled with the Allplex™-nCoV Assay (Seegene Inc., Seoul, Korea). The two methods (AusDiagnostics and Alplex^TM^-nCoV Assay SARS-CoV-2) had 98.6% agreement with the reference method, cobas 6800, for the detection of SARS-CoV-2. Agreement between the AusDiagnostics assay and the Alplex^TM^ RV Essential Assay for the detection of seven respiratory viruses was 99%. In our experience, the Respiratory Viruses 16-Well Assay proved to be the most valuable and useful medium-throughput method for simultaneous detection of important respiratory viruses and SARS-CoV-2. The main advantages of the method are high specificity for all targets included and their simultaneous detection and medium throughput with the option of having multiple instruments provide a constant run.

## 1. Introduction

Severe acute respiratory syndrome coronavirus 2 (SARS-CoV-2), which causes coronavirus disease 2019 (COVID-2019), was discovered in December 2019 in Hubei Province, China [1]. The newly discovered virus belongs to the *Betacoronavirus* B lineage and at that time had 80% similarity to the genome of severe acute respiratory syndrome virus (SARS-CoV), 50% to Middle East respiratory syndrome virus (MERS-CoV), and 96% to bat coronavirus RaTG13 [1,2]. The clinical symptoms caused by SARS-CoV-2 are nonspecific and can be associated with most other respiratory viruses that are also responsible for acute respiratory infections (ARI). The most common clinical symptoms of SARS-CoV-2 infections are fever (83–98.6%), cough (47–76%), dyspnea (14–55%), myalgia or fatigue (31–44%), and diarrhea (2–10.1%) [3,4]. With the emergence of the new genomic variant of SARS-CoV-2, Omicron, and its lower affinity for lung cells and resulting in milder illness [5,6], clinical symptoms became more ambiguous and resembled those of other respiratory viruses. Therefore, the clinical diagnosis and differentiation of COVID-19 from non-COVID-19 patients are difficult.

In addition, with the lifting of all COVID-19 restrictions, such as social distancing, lockdown, and mandatory masks in schools and in public, other respiratory viruses are expected to re-emerge in the upcoming 2022/2023 respiratory season. Because COVID-19 patients are treated differently (quarantine, isolation, and treatment), distinguishing between the virus infections will likely be more important than ever.

Since 30 January 2020, when the World Health Organization (WHO) declared an epidemic and COVID-19 a Public Health Emergency of International Concern (PHEIC), the need for commercially available tests for the reliable laboratory diagnosis of COVID-19 has increased tremendously (https://www.worldometers.info/coronavirus/, accessed on 12 July 2022). Currently, molecular assays are the most sensitive and specific methods available for case identification, control, and prevention of viral spread [7,8,9].

Simultaneous detection of respiratory viruses, including SARS-CoV-2, is necessary because the early clinical symptoms caused by SARS-CoV-19 overlap with symptoms caused by other seasonal respiratory viruses. Furthermore, the number of samples intended for laboratory evaluation increased 30-fold, from 100 to thousands of samples per day. Therefore, to ensure a reasonable turnaround time, it was necessary to switch to semi- or fully automated methods for the detection of significant respiratory viruses along with the specific detection of SARS-CoV-2 [10].

This study compares two broadly used commercial methods for the detection of respiratory viruses with the newly added target for SARS-CoV-2: the Respiratory Viruses 16-Well Assay (AusDiagnostics Pty Ltd., Mascot, Australia) Version 19 (V.19) and the Allplex™ RV Essential Assay/Allplex™-nCoV Assay (Seegene Inc., Seoul, Korea). To evaluate the performance of SARS-CoV-2 in the combinatory assays, we compared both methods with the SARS-CoV-2 kit on cobas 6800 (Roche Molecular Systems, Branchburg, NJ, USA), which was used as a reference method because it had been thoroughly evaluated previously [10].

## 2. Materials and Methods

### 2.1. Specimen Selection, Collection and Routine Diagnostics

For the study, we selected a total of 371 clinical specimens of 371 patients (Figure 1): 260 for the validation of the Respiratory Viruses 16-Well Assay (AusDiagnostics, Pty Ltd., Mascot, Australia) and an additional 111 specimens for the validation with all three methods (altogether 293 specimens, of these 147 SARS-CoV-2-positives and 146 SARS-CoV-2-negatives). All specimens were collected between 31 January and 31 March 2021, and they were initially tested for selected respiratory viruses using routine methods and were retrospectively selected for the study. Nasopharyngeal or oropharyngeal swabs (NP or OP swabs) were collected using flocked-tip swabs and transported in the Universal Transport Medium (UTM-RT) system (Copan Italia, Brescia, Italy) to the Institute of Microbiology and Immunology, Faculty of Medicine, University of Ljubljana.

The study protocol complied with the Declaration of Helsinki, the Oviedo Convention on Human Rights and Biomedicine, and the Slovenian Code on Medical Deontology, and it was approved by the Medical Ethics Committee of the Republic of Slovenia (No. 0120-211/2020/7). All data were linked exclusively to randomized numeric codes.

Swabs were vortexed at maximum speed for 1 min, and then nucleic acid extraction (NA) was performed automatically from 200 µL of the sample on a MagNA Pure Compact instrument (Roche Applied Science, Manheimm, Germany) using the MagNA Pure Compact Nucleic Acid Isolation Kit 1 (Roche), according to the manufacturer’s instructions. Before NA extraction, equine arteritis virus, an animal positive-sense single-stranded RNA virus, was added to all clinical samples as an internal extraction and amplification control. After extraction, the Respiratory Viruses 16-Well Assay V.17 (AusDiagnostics, Mascot, Australia) was performed to detect respiratory viruses, and remaining NA was immediately stored at −30 °C. In addition, a SARS-CoV-2 assay was performed from the remnants of the original sample on the cobas 6800 system, as previously described [10].

### 2.2. Respiratory Viruses’ Validation Panel

Based on the results of the Respiratory Viruses 16-Well Assay V.17 (AusDiagnostics, Mascot, Australia), 260 stored NA samples were used to perform the comparison between the Respiratory Viruses 16-Well Assay V.17 and V.19 (AusDiagnostics, Mascot, Australia). The new Respiratory Viruses 16-Well Assay V.19 includes the same viruses as version V.17: Flu A (H1, H3, H5, and H7), Flu B (Yamagata and Victoria lineages), RSV (types A and B), HRV (types A, B, and C), enterovirus (EV; types A, B, C and D), human bocavirus 1 (HBoV1), PIV (types 1 to 4), human parechovirus (types 1 to 8), HAdV (groups B, C and E, and some from groups A and D), and human coronaviruses (HCoVs; 229E, KHU-1, NL63, and OC43, plus HMPV types A and B) with the addition of SARS-CoV-2. The assay uses a human reference gene to control sample adequacy and amplification and does not provide a quantitative value for the pathogens in the samples. Interpretation of the test is automatic, based on predefined parameters by the manufacturer. The software described the target as Present or not detected. For the comparison between V.19 and the Allplex™ RV Essential Assay, 293 samples were used. Allplex™ RV Essential Assay reactions were set up automatically using a Microlab NIMBUS (Hamilton Robotics, Reno, NV, USA). Amplification was performed in a CFX96 Real-Time system instrument (BIO-RAD, Hercules, CA, USA) according to the manufacturer’s instructions. Results were automatically retrieved from Seegene Viewer software (Seegene, Seoul, Korea). Seven respiratory viruses are included in the Allplex™ RV Essential Assay: influenza A virus (Flu A) and influenza B virus (Flu B), respiratory syncytial virus (RSV), human metapneumovirus (HMPV), human adenovirus (HAdV), rhinovirus (HRV), human parainfluenza virus (PIV), and SARS-CoV-2.

### 2.3. SARS-CoV-2 Validation Panel

A total of 293 samples were used for detection using both methods: 147 confirmed SARS-CoV-2 samples and 146 SARS-CoV-2-negative samples. The Respiratory Viruses 16-Well Assay V.19 (AusDiagnostics, Mascot, Australia) was performed according to the manufacturer’s instructions. Two SARS-CoV-2 targets (ORF8 and ORF1) are included in the V.19 assay. Similar to the processing of samples in the respiratory viruses panel, the Allplex™-nCoV Assay was set up automatically on a Microlab NIMBUS (Hamilton Robotics, Reno, NV, USA) system, with amplification and detection performed on a CFX96 Real-Time instrument (BIO-RAD, Hercules, CA, USA) according to the manufacturer’s instructions. Interpretation was performed automatically using the Seegene Viewer Software (Seegene, Seoul, South Korea) and manually by observing the amplification curves of the E, N, and RdRP/S genes. Finally, SARS-CoV-2 test results from both assays were compared to the SARS-CoV-2 routine method on cobas 6800 (Roche, Mannheim, Germany) [10].

### 2.4. Specificity Testing

For specificity testing, 147 SARS-CoV-2-positive and 146 SARS-CoV-2-negative samples were included. In addition, NA was extracted from 147 SARS-CoV-2-positive samples, and SARS-CoV-2 RNA concentrations were quantified using the LightMix^®^ E-gene kit (Roche, Mannheim, Germany) with a synthetic DNA calibration standard (gBlock, IDT Technologies, Coralville, IA, USA). The mean SARS-CoV-2 concentration in selected samples was 4.4 × 10^4^ RNA copies/µL (min. 13 copies/µL; max. 4.5 × 10^7^ copies/µL). When the specificity was assessed, 27 of 146 SARS-CoV-2-negative samples were negative for all respiratory viruses, 59 were positive for one, 43 contained a mixture of two, 16 contained three, and 1 contained five different respiratory viruses.

### 2.5. Statistical Analysis

The results of all tests were analyzed separately in 2 × 2 tables according to the combination of tests and viral target. Method agreement, Cohen’s kappa, and the Pearson correlation coefficient between Ct values were calculated [11]. All analyses were performed using Graph Pad Prism 7 (GraphPad software) version 7.04.

## 3. Results

### 3.1. Respiratory Viruses 16-Well Assay V.17 versus V.19

The new version V.19 for respiratory virus detection was in complete agreement with the replaced assay version V.17 for all viruses tested, except for RSV (Table 1). One more positive RSV was detected with the new version V.19 than with version V.17 (kappa = 0.98; 95% CI: 0.95–1.0). The correlation in Ct values was statistically significant for all targets (α = 0.05).

### 3.2. Respiratory Viruses 16-Well Assay V.19 and Allplex™-nCoV Assay SARS-CoV-2 Detection Performance for Detection of SARS-CoV-2

Altogether, 293 samples were successfully analyzed with the Allplex™-nCoV Assay and the Respiratory Viruses 16-Well Assay V.19. Of the 147 SARS-CoV-2-positive samples previously confirmed with routine methods, 97.3% (143/147) were identified as positive with V.19. When the automated algorithm was used to call the results with the Allplex™-nCoV Assay (a sample is positive if at least one target gene is detected), the same result was obtained (143/147; 97.3%). The lowest SARS-CoV-2 RNA concentration detected was 26 copies/µL and 13 copies/µL for the Respiratory Viruses 16-Well Assay V.19 and the Allplex™-nCoV Assay, respectively. However, when a more conservative approach was used—namely, that SARS-CoV-2 is positive only when all three genes or at least two genes are detected—the Allplex™-nCoV Assay (Seegene, Seoul, South Korea) correctly identified 90.5% (133/147) and 91.8% (135/147) of the samples, respectively. The lowest concentration of SARS-CoV-2 that still resulted in detection of all three genes was 103 copies/µL and 83 copies/µL for two targets. False positive results were not observed for any of the methods tested (0/147), regardless of whether samples were previously known to be completely negative (27/146 SARS-CoV-2-negative samples) or positive for one or more non-SARS-CoV-2 respiratory viruses (119/146), including seasonal HCoVs (24/119). The percent agreement between cobas 6800, the Respiratory Viruses 16-Well Assay V.19, and the Allplex™-nCoV Assay (with automatic result retrieval) for the detection of SARS-CoV-2 was 98.6% (Cohen’s kappa index: 0.97). Using a more conservative approach, the percent agreement between the Allplex™-nCoV Assay and cobas 6800 decreased to 95.9% (Cohen’s kappa index: 0.92) and 95.2% (Cohen’s kappa index: 0.90), respectively, when at least two or all three target genes must be detected in a sample to be considered SARS-CoV-2-positive (Table 2).

A comparison between the Respiratory Viruses 16-Well Assay V.19 and the Allplex™-nCoV Assay showed 98.6% agreement (Cohen’s kappa index: 0.97) between these two methods when only one target was considered sufficient to classify a sample as SARS-CoV-2-positive using the Allplex™-nCoV Assay. When a stricter interpretation criterion was used, the agreement dropped to 96.6% when all three or two target genes had to be detected to classify a sample as SARS-CoV-2-positive by the Allplex™-nCoV Assay (Table 3).

### 3.3. Comparison of the Respiratory Viruses 16-Well Assay V.19 and the Allplex™ RV Essential Assay

In total 293 samples with none, one, or a combination of different respiratory viruses (including SARS-CoV-2) were successfully tested using both methods. All 27 samples with previously undetected viral pathogens were also negative by both methods. Because the Allplex™ RV Essential Assay detects fewer viral targets than the Respiratory Viruses 16-Well Assay V.19, only the overlapping viruses were included in the method comparison. For all viruses tested (FluA, FluB, PIV, HMPV, RSV, HAdV) except HRV, agreement was greater than 99%. For HRV, the agreement was 95.6% (detailed results in Table 4).

### 3.4. Cross Reactivity of SARS-CoV-2 with Seasonal HCoVs in the Respiratory Viruses 16-Well Assay V.19

Among the 293 samples successfully tested by all three methods (the Respiratory Viruses 16-Well Assay V.19, cobas 6800, and the Allplex™-nCoV Assay), seasonal HCoVs were detected in 15% (44/293) of the samples. Twenty of these were also positive for SARS-CoV-2, as expected from routine results. Seasonal HCoVs as the only pathogen were detected in 24 samples. When seasonal HCoVs and SARS-CoV-2 were detected simultaneously by the Respiratory Viruses 16-Well Assay V.19, the median Ct-value for seasonal HCoVs was higher (median Ct-value 29.7; range 20.6–33.8) than the median Ct-value for SARS-CoV-2 (median Ct-value 11.3; range 8.7–20.4). When seasonal HCoVs were the only pathogen detected in samples, the median Ct-value was significantly lower, at 19.9; range 11.7–30.4 *(p* < 0.0001).

## 4. Discussion

Respiratory viruses are an important cause of human morbidity and mortality worldwide and are a major global health problem. Since the emergence of SARS-CoV-2 in December 2019, the virus has been recognized as the cause of more than 516 million infections and more than 6 million deaths worldwide (https://www.worldometers.info/coronavirus/, accessed on 12 July 2022) [1]. Because the clinical signs of SARS-CoV-2 infection overlap with the signs and symptoms of other respiratory viral infections, commercially available tests to detect all respiratory viruses, including SARS-CoV-2, are needed to help clinicians isolate infectious patients and organize clinical care. At the onset of the SARS-CoV-2 epidemic in the 2019/2020 respiratory season, a very low number of respiratory coinfections, including influenza virus with SARS-CoV-2, were detected [12,13,14,15]. Restriction implementation, lockdowns, and mandatory masks may affect the transmission and spread of all respiratory viruses, including influenza virus, and thus the low coinfection outcomes. With the emergence of the SARS-CoV-2 Omicron genetic variant, which causes a milder disease [5,6] and symptoms resembling other respiratory infections, differentiation among respiratory viruses is becoming increasingly important. The AusDiagnostics platform with the Respiratory Viruses 16-Well Assay V.17, which was already used in our laboratory, was upgraded to a new version (V.18), in which the SARS-CoV-2 (ORF1) target was added. In the final version (V.19) evaluated in our study, an additional SARS-CoV-2 target was added (ORF 1 and ORF8). Both methods (V.17 and V.19) showed complete agreement for all viruses tested, except RSV (one additional positive RSV was detected with V.19 compared to V.17), leading to the conclusion that the addition of SARS-CoV-2 to the panel does not affect performance in detecting other respiratory viruses. The discrepancy of RSV detection was probably because of a low viral load in sample material (Ct-value was 33.3) or better sensitivity of new version V.19 compared to V.17 for RSV. The positive results were also confirmed with Xpert Xpress Flu/RSV assay (Cepheid, Sunnyvale, CA, USA) but data were not shown. In the present study one sample with five detected viruses was included. Multiple detections of different respiratory viruses are common, especially in small children and immunocompromised patients receiving immunosuppressing therapy [16]. At the point of testing, targets can be detected as the results of recovery of infection or as the results of acute infection.

The AusDiagnostics multiplex tandem PCR (MT-PCR) V.19 assay was additionally compared to the Allplex™ RV Essential Assay for the detection of seven respiratory viruses (FluA, FluB, PIV, HMPV, RSV, AdV, and HRV) to better estimate the performance of the AusDiagnostics kit. The concordance between the two methods was 99% for all overlapping viruses, except for HRV (95.6%). Because HRV is closely related to EV and differentiation between these two viruses is notoriously difficult, this result is not unexpected. However, the major limitation of the Allplex™ RV Essential Assay is that not all clinically relevant respiratory viruses are included in one assay.

Regarding SARS-CoV-2 testing, all three methods (Respiratory Viruses 16-Well Assay V.19, Allplex™-nCoV Assay SARS-CoV-2, and cobas 6800) had 98.6% agreement (using Allplex automatic results retrieval). Overall agreement between Allplex™-nCoV Assay and cobas 6800 decreased from 95.9% to 95.2%, respectively, when two or all three genes had to be detected to consider a sample SARS-CoV-2-positive. The same concordance was observed between the Respiratory Virus 16-Well Assay V.19 and the Allplex™ RV Essential Assay for detection of SARS-CoV-2 (98.6%), when at least one target gene was detected by Allplex, and lower when two or all three target genes were detected (96.6%). We advise caution in interpreting the results of the Seegene Viewer Software because it designates a sample as SARS-CoV-2-positive even when only one target gene is positive with a very high Ct value (>36). Moreover, the interpretation of results with Allplex™-nCoV Assay SARS-CoV-2 is less reliable if only one or two out of three genes are detected. Another difference between these two observed methods is in using different target numbers for the detection of SARS-CoV-2. Respiratory Viruses 16-Well Assay V.19 uses two target genes (ORF8, ORF1) while the Allplex™-nCoV Assay SARS-CoV-2 observed the amplification curves of three genes (E, N, and RdRP/S). With the emergence of new variants of the virus SARS-CoV-2, this could be a problem. When the present study was performed (from 31 January to 31 March 2021) genomic variant Omicron B.a.5., which is now rapidly replacing the previous B.a.2., had not emerged yet. However, according to the results of the research carried out by Kogoj et al. [17], where ten clinically most relevant SARS-CoV-2 genomic variants were compared with six different diagnostic approaches. The differences in Ct-values between different genomic variants and platforms were observed, thus close monitoring of new emerging SARS-CoV-2 genomic variants is needed. Until now, different Omicron genetic variants have not affected the detection of SARS-CoV-2 (data not shown).

Comparing all three platforms in terms of throughput, ease of use, and turnaround time is rather difficult due to the differences in system design and purpose. When we considered only the Respiratory Viruses 16-Well Assay V.19 and the Allplex™ RV Essential Assay in combination with the Allplex™-nCoV Assay, we found the Respiratory Viruses 16-Well Assay V.19 to be more user friendly. Furthermore, the Respiratory Viruses 16-Well Assay V.19 includes a broader spectrum of respiratory viruses and is therefore more useful than the combination of both Allplex assays. However, an important limitation of the Respiratory Viruses 16-Well Assay V.19, which has to be kept in mind, is that it is not suitable for mass testing because a maximum of 22 samples can be tested in one run. In addition, the turn-around time for the Respiratory Viruses 16-Well Assay V.19, along with nucleic acid extraction, is approximately 4 h, which is longer compared to cobas 6800, which requires 3 h for 94 samples. However, the Allplex assay requires 5 h per run, and when both are loaded together on a MicroLab NIMBUS (48-sample version) only a maximum of 22 samples can be processed. However, a possible solution to the lower throughput of the AusDiagnostics platform, if available, would be to have multiple instruments to provide a constant run. Nonetheless, AusDiagnostics MT-PCR is the faster commercially available automated system that includes all respiratory viruses in one assay and allows simultaneous processing of more samples, which is not the case for FilmArray (BioFire Diagnostics, LLC, Salt Lake City, UT, USA), for example. In a study from Australia, in which a large number of clinical samples (7839) were tested, the Respiratory Viruses 16-Well Assay V.19 was shown to be a reliable tool for the detection of SARS-Cov-2 [18]. Only 7.1% of the tests had discordant results. All were SARS-CoV-2-positives with the AusDiagnostics assay and negative with an in-house real-time TaqMan PCR assay performed in a reference laboratory [19]. In our study, only 4/143 (2.8%) were negative with AusDiagnostics and positive with cobas 6800. On the other hand, 20/143 samples showed positive signals for both seasonal HCoVs and SARS-CoV-2, and the median Ct values were much higher for seasonal HCoVs than for SARS-CoVs (29.7 vs. 11.3). This observation, combined with the fact that all these samples were cobas 6800-confirmed SARS-CoV-2 cases, suggests the possibility of cross-reactivity, particularly if the samples contain a high concentration of SARS-CoV-2. In contrast, we did not observe any problems with such cross-reactivity in the opposite combination (high Ct-value for SARS-CoV-2 and low Ct-value for HCoVs). Similar results were also observed by Rahman et al. [20]. Finally, an important feature of the AusDiagnostics method is the ability to verify the quality of swab collection by assessing the amount of the human reference gene used for amplification control, which allows better interpretation of results with low target concentrations.

## 5. Conclusions

In conclusion, because of the lifting of all COVID-19 restriction measures all over the world and the immediate emergence of other respiratory viruses, including the influenza A virus, we believe that the combinatory diagnostics of different respiratory viruses, including SARS-CoV-2, are needed to control isolation and manage hospital care. In our study we compared two options: first, the use of the AusDiagnostics Respiratory Viruses 16-Well Assay V.19 with medium throughput but the simultaneous detection of 16 respiratory viruses, including SARS-CoV-2, and second, a combination of the Allplex™ RV Essential Assay with Allplex™-nCoV, which has automatic isolation and a medium throughput and includes seven clinically important respiratory viruses. In our experience, we believe the AusDiagnostics Respiratory Viruses 16-Well Assay V.19 to be the most valuable and useful medium throughput method, with the main advantage of the simultaneous detection of all respiratory viruses, including SARS-CoV-2. The main advantages of the method are the high specificity for all targets included and their simultaneous detection and medium throughput with the option to have multiple instruments to provide a constant run.

## Figures and Tables

**Figure 1 viruses-14-01530-f001:**
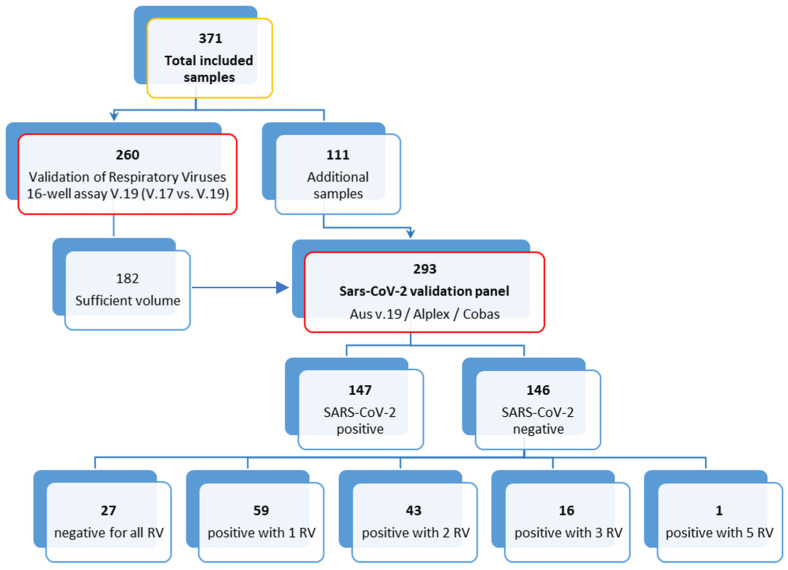
Flow chart of the specimens included in the study. RV = respiratory viruses.

**Table 1 viruses-14-01530-t001:** Comparison of the Respiratory Viruses 16-Well Assay V.17 versus V.19 for respective targets.

V.19 for respiratory viruses	**V.17 for Respiratory Viruses**				**% Agreement**	**Kappa**	** *r* **
			**Pos**	**Neg**			
	FluA	Pos	22	0	100 (100–100)	1 (1–1)	0.95
		Neg	0	238			
	FluB	Pos	24	0	100 (100–100)	1 (1–1)	0.70
		Neg	0	236			
	RSV	Pos	35	1	99.6 (97.9–99.9)	0.98 (0.95–1)	0.70
		Neg	0	224			
	HRV	Pos	52	0	100 (100–100)	1 (1–1)	0.98
		Neg	0	208			
	EV	Pos	9	0	100 (100–100)	1 (1–1)	0.99
		Neg	0	251			
	Parecho	Pos	8	0	100 (100–100)	1 (1–1)	0.98
		Neg	0	252			
	HBoV	Pos	28	0	100 (100–100)	1 (1–1)	0.99
		Neg	0	232			
	PIV	Pos	20	0	100 (100–100)	1 (1–1)	0.99
		Neg	0	240			
	AdV	Pos	22	0	100 (100–100)	1 (1–1)	0.99
		Neg	0	238			
	HMPV	Pos	22	0	100 (100–100)	1 (1–1)	0.99
		Neg	0	238			
	HCoV	Pos	25	0	100 (100–100)	1 (1–1)	0.96
		Neg	0	235			

Data are shown as 95% confidence interval for % agreement and Kappa.

**Table 2 viruses-14-01530-t002:** Comparison between cobas 6800 (reference method), the Respiratory Viruses 16-Well Assay V.19, and the Allplex™-nCoV Assay.

		Cobas 6800		% Agreement	Kappa
		Pos	Neg		
V.19 for respiratory viruses (AusDiagnostics)	Pos	143	0	98.6 (96.5–99.5)	0.97 (0.94–1.00)
	Neg	4	146		
Allplex™-nCoV Assay (automatic calling)	Pos	143	0	98.6 (96.5–99.5)	0.97 (0.94–1.00)
	Neg	4	146		
Allplex™-nCoV Assay (all 3 genes for pos)	Pos	133	0	95.2 (92.1–97.1)	0.90 (0.85–0.95)
	Neg	14	146		
Allplex™-nCoV Assay (at least 2 genes for pos)	Pos	135	2	95.9 (93.0–97.6)	0.92 (0.87–0.96)
	Neg	10	146		

Data are shown as 95% confidence interval for % agreement and Kappa.

**Table 3 viruses-14-01530-t003:** Comparison between the Respiratory Viruses 16-Well Assay V.19 and the Allplex™-nCoV Assay.

	V.19 for Respiratory Viruses (AusDiagnostics)			% Agreement	Kappa
		Pos	Neg		
Allplex™-nCoV Assay (automatic calling)	Pos	141	2	98.6 (96.5–99.5)	0.97 (0.95–1.00)
	Neg	2	148		
Allplex™-nCoV Assay (all 3 genes for pos)	Pos	133	0	96.6 (93.8–98.1)	0.93 (0.89–0.97)
	Neg	10	150		
Allplex™-nCoV Assay (at least 2 genes for pos)	Pos	135	2	96.6 (93.8–98.1)	0.93 (0.89–0.97)
	Neg	8	148		

Data are shown as 95% confidence interval for % agreement and Kappa.

**Table 4 viruses-14-01530-t004:** Results comparison between the Respiratory Viruses 16-Well Assay V.19 and the Allplex™ RV Essential Assay for influenza A (FluA), influenza B (FluB), parainfluenza (PIV), respiratory syncytial virus (RSV), metapneumovirus (MPV), respiratory strains of adenovirus (AdV), and human rhinovirus (HRV).

		V.19 for Respiratory Viruses (AusDiagnostics)													
		FluA		FluB		PIV		RSV		MPV		AdV		HRV	
		Pos	Neg	Pos	Neg	Pos	Neg	Pos	Neg	Pos	Neg	Pos	Neg	Pos	Neg
Seegene	Pos	14	0	14	1	7	0	24	2	12	2	22	3	32	2
	Neg	1	278	0	278	3	283	1	266	1	278	0	268	11	248
% agreement		99.7 (98.1–99.9)		99.7 (98.1–99.9)		99.0 (97.0–99.7)		99.0 (97.0–99.7)		99.0 (97.0–99.7)		99.0 (97.0–99.7)		95.6 (92.6–97.4)	
Kappa		0.96 (0.89–1.00)		0.96 (0.89–1.00)		0.82 (0.62–1.00)		0.94 (0.86–1.00)		0.88 (0.75–1.00)		0.93 (0.85–1.00)		0.81 (0.70–0.91)	

Data are shown as 95% confidence interval for % agreement and Kappa.

## Data Availability

Not applicable.

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
