# Peer review of "Evaluation of Two Broadly Used Commercial Methods for Detection of Respiratory Viruses with a Recently Added New Target for Detection of SARS-CoV-2"

_viruses, 2022, doi:10.3390/v14071530_

Round 1

Reviewer 1 Report

Viruses (ISSN 1999-4915)

Manuscript ID: viruses-1760915

Evaluation of two broadly used commercial methods for detection of respiratory viruses with a recently added new target for detection of SARS-CoV-2

Monika Jevšnik Virant , Tina Uršič , Rok Kogoj , Miša Korva , Miroslav Petrovec , Tatjana Avšič Županc 

Very interesting and well written paper on an important topic. However, I have some comments and suggestions which may improve the quality of this manuscript.

We are in the middle of a new wave of Covid. It has emerged that in many countries there is a significant increase in the sub-variant Omicron B.a.5, which is rapidly replacing the previous B.a.2. Do the current methods for detection of respiratory viruses, evaluated in this study, have the same sensitivity and specificity for the emerging Omicron variants? It would be helpful to add a short paragraph in the Discussion.

In all tables it is appropriate to add 95% Confidence Interval for % agreement and Kappa.

Author Response

Reviewer 1

Ad1. Reviewer 1

Very interesting and well written paper on an important topic. However, I have some comments and suggestions which may improve the quality of this manuscript.

Ad1. Authors’ response

We would like to thank the Reviewer for this general positive comment about our manuscript.

Ad2. Reviewer 1

We are in the middle of a new wave of Covid. It has emerged that in many countries there is a significant increase in the sub-variant Omicron B.a.5, which is rapidly replacing the previous B.a.2. Do the current methods for detection of respiratory viruses, evaluated in this study, have the same sensitivity and specificity for the emerging Omicron variants? It would be helpful to add a short paragraph in the Discussion.

Ad2. Authors’ response

The following sentences have been included in a revised version of the manuscript in the Discussion section: page 10 line 299  “With the emergence of new variants of the virus SARS-CoV-2, this could be a problem. When the present study was performed (from January 31st to March 31st, 2021) genomic variant Omicron B.a.5. which is now rapidly replacing the previous B.a.2. had not emerged yet.  However, according to the results of the research done by Kogoj at all., where 10 clinically most relevant SARS-CoV-2 genomic variants were compared with 6 different diagnostic approaches. The differences in Ct-values between different genomic variants and platforms were observed, thus close monitoring of new emerging SARS-CoV-2 genomic variants is needed. Till now different Omicon genetic variants did not affect the detection of SARS-CoV-2 (data were not shown).”

New reference has been included: Kogoj R, Korva M, Knap N, Resman Rus K, Pozvek P, Avsic-Zupanc T, et al. Comparative Evaluation of Six SARS-CoV-2 Real-Time RT-PCR Diagnostic Approaches Shows Substantial Genomic Variant-Dependent Intra- and Inter-Test Variability, Poor Interchangeability of Cycle Threshold and Complementary Turn-Around Times. Pathogens. 2022;11(4).   

Ad3. Reviewer 1

In all tables it is appropriate to add 95% Confidence Interval for % agreement and Kappa.

Ad3. Authors’ response

Thank you for your observation. 95 % Confidence Interval for % agreement and Kappa were added to all tables (1-4).

Reviewer 2 Report

With great interest I read the manuscript of Jevšnik Virant et al, which compares and also evaluates two broadly used commercial methods for detection of respiratory viruses with a recently added novel target, that of SARS-CoV-2.

I find the manuscript interesting, well written and the content might also be proven useful in the real-world settings especially for the upcoming fall/winter period where respiratory viral infections are on their peak, their clinical phenotypes similar and the need for easy molecular identification imperative. However, I as well as the authors should point out that most cases of Covid-19 have been diagnosed by single PCR or antigen tests that are readily available and not by multiplex assays. Even though, the manuscript merits publication after some minor additions necessary to improve its overall quality.

Firstly it could be clarified whether the detection of one gene –out of three- with the Allplex- nCoV assay, is significantly less reliable for the characterization of a sample as SARS-COV-2 positive. In line 119 is implicated that Respiratory Viruses 117 16-Well Assay V.19, uses 2 target genes (ORF8,ORF1) for the detection of SARS-COV-2 while the Allplex-nCoV assay observes the amplification curves of up to 3 genes (E, N, and RdRP/S). This difference between the 2 methods could be clearly stated.

Could you please explain why not all the 260 samples used for the validation of the 16-Well Assay were not used for the head-to-head comparison of the 3 methods. On what ground were the 149 used for both parts of the study samples selected?

Please elaborate further the “statistical methods” part of the manuscript. Please included information as the program used for the comparisons, the definition of statistical importance, Cohen’s kappa coefficient threshold for a comparison to be considered as almost in perfect agreement etc.

It would be nice if you could mention which Ct value you considered the limit of positivity (was it 40?). What was the agreement in the Ct values between the different methods of detection?

Please accompany median with the range of values.

To facilitate the reader, could you please share the writers’ opinion on the detection of 5 different viruses in the same patient and the difference in the detection of an RSV case between version V.17 and 19 of the 16-Well Assay.

It would be nice if you could also mention another important aspect of the methods used, the involved cost.

Author Response

Reviewer 2

Ad1. Reviewer 2

With great interest I read the manuscript of Jevšnik Virant et al, which compares and also evaluates two broadly used commercial methods for detection of respiratory viruses with a recently added novel target, that of SARS-CoV-2.

I find the manuscript interesting, well written and the content might also be proven useful in the real-world settings especially for the upcoming fall/winter period where respiratory viral infections are on their peak, their clinical phenotypes similar and the need for easy molecular identification imperative. However, I as well as the authors should point out that most cases of Covid-19 have been diagnosed by single PCR or antigen tests that are readily available and not by multiplex assays. Even though, the manuscript merits publication after some minor additions necessary to improve its overall quality.

Ad1. Authors’ response

We would like to thank the Reviewer for these valued comments about our manuscript.

Ad2. Reviewer 2

Firstly it could be clarified whether the detection of one gene –out of three- with the Allplex- nCoV assay, is significantly less reliable for the characterization of a sample as SARS-COV-2 positive. In line 119 is implicated that Respiratory Viruses 117 16-Well Assay V.19, uses 2 target genes (ORF8,ORF1) for the detection of SARS-COV-2 while the Allplex-nCoV assay observes the amplification curves of up to 3 genes (E, N, and RdRP/S). This difference between the 2 methods could be clearly stated.

Ad2. Authors’ response

The following sentences have been included in a revised version of the manuscript in the Discussion section: page 10 line 295 “Moreover, interpretation of results with Allplex™ ‑nCoV Assay SARS ‑CoV-2 is less reliable if only one or two out of three genes are detected. Another difference between these two observed methods is in using different target numbers for the detection of SARS-CoV-2. Respiratory Viruses 16-Well Assay V.19 uses two target genes (ORF8, ORF1) while the Allplex™ ‑nCoV Assay SARS ‑CoV-2 observed the amplification curves of three genes (E, N, and RdRP/S).”

Ad3. Reviewer 2

Could you please explain why not all the 260 samples used for the validation of the 16-Well Assay were not used for the head-to-head comparison of the 3 methods. On what ground were the 149 used for both parts of the study samples selected?

Ad3. Authors’ response

Thank you for your observation. In accordance with the Reviewer’s suggestion, the following sentence has been completed in the corrected version of the manuscript: page 2 line 78: “ …(altogether 293 specimens, of this 147 SARS-CoV-2 positive and 146 SARS-CoV-2 negatives).“

For better understanding of the included and tested samples, Figure 1 has been formatted. All 260 samples could not be used for further head-to-head comparison analysis, due to insufficient volume of 111 included samples.  (Figure 1). For better comparison and validation of SARS-CoV-2, approximately half of the 293 included samples were SARS-CoV-2 positive and half were negative.

Ad4. Reviewer 2

Please elaborate further the “statistical methods” part of the manuscript. Please included information as the program used for the comparisons, the definition of statistical importance, Cohen’s kappa coefficient threshold for a comparison to be considered as almost in perfect agreement etc.

Ad4. Authors’ response

Page 5, line 154: The information about the program used for the comparisons has been added: “All analyses were performed using Graph Pad Prism 7 (GraphPad software) version 7.04.” The following reference has been included on page 5 line 157: 1. Landis, J.R.; Koch, G.G. (1977). The measurement of observer agreement for categorical data. Biometrics. 33 (1): 159-174. https://doi.org/10.2307%2F2529310, where an almost perfect agreement for Cohen’s kappa coefficient was between 0.81 and 1.00.

Ad5. Reviewer 2

It would be nice if you could mention which Ct value you considered the limit of positivity (was it 40?). What was the agreement in the Ct values between the different methods of detection?

Ad5. Authors’ response

We would like to thank the Reviewer for this suggestion, but unfortunately information about Ct value that was considered for the limit of positivity we can not give. Consideration about positivity was analytical and totally automatic. Explanation about this for the Respiratory Viruses 16-Well Assay V.17 was added on page 4 line 115: “The assay uses a human reference gene to control sample adequacy and amplification and does not provide a quantitative value for the pathogens in the samples. Interpretation of the test is automatic, based on predefined parameters by the manufacturer. The software was called the target as Present or not detected.“ Interpretation of Alplex results has already been given on page 4 lines 122 and 138. Comparison of Ct-value between different assays is complex and depends on different factors. For better understanding following sentences were added in the Discussion compartment: on page 10 line 295 ““Moreover, interpretation of results with Allplex™ ‑nCoV Assay SARS ‑CoV-2 is less reliable if only one or two out of three genes are detected. Another difference between these two observed methods is in using different target numbers for the detection of SARS-CoV-2. Respiratory Viruses 16-Well Assay V.19 uses two target genes (ORF8, ORF1) while the Allplex™ ‑nCoV Assay SARS ‑CoV-2 observed the amplification curves of three genes (E, N, and RdRP/S). With the emergence of new variants of the virus SARS-CoV-2, this could be a problem. When the present study was performed (from January 31st to March 31st, 2021) genomic variant Omicron B.a.5. which is now rapidly replacing the previous B.a.2. had not emerged yet.  However, according to the results of the research done by Kogoj at all., where 10 clinically most relevant SARS-CoV-2 genomic variants were compared with 6 different diagnostic approaches. The differences in Ct-values between different genomic variants and platforms were observed, thus close monitoring of new emerging SARS-CoV-2 genomic variants is needed. Till now different Omicon genetic variants did not affect the detection of SARS-CoV-2 (data were not shown).”

New reference has been included: Kogoj R, Korva M, Knap N, Resman Rus K, Pozvek P, Avsic-Zupanc T, et al. Comparative Evaluation of Six SARS-CoV-2 Real-Time RT-PCR Diagnostic Approaches Shows Substantial Genomic Variant-Dependent Intra- and Inter-Test Variability, Poor Interchangeability of Cycle Threshold and Complementary Turn-Around Times. Pathogens. 2022;11(4).   

Ad6. Reviewer 2

Please accompany median with the range of values.

Ad6. Authors’ response

The range of values from median Ct values has been added in compartment 3.4 page 9 line 241.

Ad7. Reviewer 2

To facilitate the reader, could you please share the writers’ opinion on the detection of 5 different viruses in the same patient and the difference in the detection of an RSV case between versions V.17 and 19 of the 16-Well Assay.

Ad7. Authors’ response

The following sentences have been included in a revised version of the manuscript in the Discussion section: on page 9 line 268 “Discrepancy of RSV detection was probably because of low viral load in sample material (Ct- value was 33.3) or better sensitivity of new version V. 19 compare to V17 for RSV. The positive results were also confirmed with GenXpert assay (Cepheid, USA) but data were not shown. In the present study, one sample with five detected viruses was included. Multiple detections of different respiratory viruses are common, especially in small children and immunocompromised patients receiving immunosuppressing therapy. At the point of testing, targets can be detected as results of recovery of infection, another as results of acute infection.” One reference has been added: page 9 line 274

Kaplan SL. Coinfections in Hospitalized Children With Community-Acquired Pneumonia: What Does This Mean for the Clinician? The Journal of infectious diseases. 2018;218(2):173-5.

Ad8. Reviewer 2

It would be nice if you could also mention another important aspect of the methods used, the involved cost.

Ad8. Authors’ response

Thank you for your excellent point of comment, but unfortunately, the prices of assays and the platforms are different across different countries due to differences in distributions, due to differences in payment and test purchase policies and differences in the number of tests performed and bought, therefore we can not comment on this subject.
